# Differential Metabolomics Reveals Pathogenesis of *Pestalotiopsis kenyana* Causing Leaf Spot Disease of *Zanthoxylum schinifolium*

**DOI:** 10.3390/jof8111208

**Published:** 2022-11-15

**Authors:** Chang Liu, Haiyao Guo, Han Liu, Jiawen Yu, Shuying Li, Tianhui Zhu, Adjei Mark Owusu, Shujiang Li

**Affiliations:** 1College of Forestry, Sichuan Agricultural University, Chengdu 611130, China; 2Ganzi Institute of Forestry Research, Kangding 626700, China; 3National Forestry and Grassland Administration, Key Laboratory of Forest Resources Conservation and Ecological Safety on the Upper Reaches of the Yangtze River & Forestry Ecological Engineering in the Upper Reaches of the Yangtze River, Key Laboratory of Sichuan Province, Chengdu 611130, China

**Keywords:** *Zanthoxylum schinifolium*, *Pestalotiopsis kenyana*, differential metabolites, LC–MS

## Abstract

Pepper leaf spot is a common disease of *Zanthoxylum schinifolium*. When it is serious, it directly affects the growth of *Z. schinifolium*, making the plant unable to blossom and bear fruit, which seriously restricts the development of the *Z. schinifolium* industry. Therefore, the pathogenic mechanism of leaf spots should be explored to provide a basis for a comprehensive understanding of the disease. Using liquid chromatography–mass spectrometry (LC–MS) technology combined with the data-dependent acquisition, the full spectrum analysis of pathogen mycelium samples was carried out. Partial least squares discriminant analysis (PLS-DA) was used to reveal the differences in metabolic patterns among different groups. Hierarchical clustering analysis (HCA) and PLS-DA were used to reveal the relationship between samples and metabolites, which reflected the metabolomics changes of *Pestalotiopsis kenyana* in the logarithmic growth phase of mycelia, the stable growth phase of mycelia, the massive spore stage, the induction culture conditions of PDA and *Z. schinifolium* leaves, and the possible pathogenic substances were selected for pathogenicity detection. PLS-DA had a strong predictive ability, indicating a clear analysis trend between different groups. The results of the metabolomics analysis showed that the differential metabolites of pathogenic bacteria were abundant at different stages and under different medium conditions, and the content of metabolites changed significantly. There were 3922 differential metabolites in nine groups under positive and negative ion modes, including lipids and lipid molecules, organic acids and their derivatives, organic heterocyclic compounds, organic oxygen compounds, carbohydrate polyketides, nucleosides, nucleotides, and analogs. The results of the pathogenicity test showed that the leaves treated with 3,5-dimethoxy benzoic acid, *S*-(5-adenosy)-l-homocysteine, 2-(1*H*-indol-3-yl) acetic acid, l-glutamic acid, and 2-(2-acetyl-3,5-dihydroxy phenyl) acetic acid showed different degrees of yellowish-brown lesions. This indicated that these substances may be related to the pathogenicity of *P. kenyana*, and the incidence was more serious when treated with 3,5-dimethoxybenzoic acid and *S*-(5-adenosy)- l -homocysteine. This study provides a basis for further analysis of differential metabolites and provides a theoretical reference for the prevention and treatment of *Z. schinifolium* leaf spot.

## 1. Introduction

*Zanthoxylum schinifolium* is a deciduous small tree of Rutaceae, *Zanthoxy* L. [1], with about 250 species native to China, mainly planted in Sichuan, Shaanxi, Gansu, Yunnan, Shandong, Guizhou, and other provinces [2,3,4], and widely distributed in Asian countries [5]. It has a long history of cultivation; its peel has a unique aroma and taste, which is widely used as a spice in food processing and cooking [6,7,8,9,10,11]. However, the incidence of *Z. schinifolium* leaf spot disease is serious, which restricts the development of the pepper industry. Since 2018, *Z. schinifolium* in Jianyang City has developed leaf spot disease, with 50% of plants showing disease symptoms [12]. At the beginning of the occurrence, yellow-brown lesions formed on the leaves; in the later stages, the area of the lesions expanded. At the severe stage, multiple lesions merged into one large, dead spot, and the plants failed to blossom and bear fruit. Therefore, exploring the pathogenic mechanism of leaf spots provides a basis for a comprehensive understanding of the disease.

Metabolomics is one of the main means of biological phenotype research and an indispensable part of system biology research. It mainly considers primary metabolites such as sugars, lipids, nucleotides, amino acids, and secondary metabolites such as terpenoids, polyketides, non-ribosomal peptides, and alkaloids [13]. Non-targeted metabolomics enables the comprehensive and unbiased metabolite analysis of metabolites in organisms with a wide range of substance coverage due to a large amount of information detected [14]. At present, metabolomics has been extended to various fields, including animal, plant, microorganism, clinical, pharmaceutical, and environmental studies [15]. In recent years, due to the importance of plant pathogenic fungi in microbial systems, metabolomics technology has been widely used in the study of plant pathogenic fungi [16], mainly focusing on fungal classification [17,18], pathogenesis, and interaction with plants in the process of metabolite function and metabolic pathways [19,20,21]. For example, the differences in metabolites between four different Fusarium strains showed that the effect of the nutritional environment on fungal metabolism was greater than that of genotype [22]; chitosan plays an important role in the formation of *S.nodorum* spores [23]. When *Ustilago maydis* induced the formation of corn tumors, the metabolites changed significantly, and its flavonoid pathway and shikimic acid pathway were activated [24]. The contents of the related metabolites phenyl propionic acid, tyrosine, and shikimic acid increased significantly, and the levels of hydroxycinnamic acid (HCA) derivatives and anthocyanins increased [24]. In the study of plant diseases, metabolomics provides important clues for the study of molecular mechanisms such as pathogen development, adaptation to environmental stress, and pathogenesis [25,26]. Therefore, it is very important to explore the pathogenic metabolites by using the changes in the nutritional environment of the *Z. schinifolium* leaf spot pathogen.

Our previous study showed that the pathogen causing *Z. schinifolium* leaf spot was *P. kenyana* [12]. *Pestalotiopsis* is a species-rich asexual genus that produces conidia and is widely distributed in tropical and temperate regions. It is a common plant pathogen and can cause a variety of diseases, including canker, shoot blight, leaf spot, gray spot, leaf blight, canker, and fruit rot [27,28,29,30,31,32,33,34]. The pathogens of banana leaf wilt and banana leaf blight are *P. micropore* and *P. theae,* respectively [35,36]. Meanwhile, *Pestalotiopsis* is also an endophytic fungus with abundant products and has become the main source of many active metabolites [37,38]. Five compounds, 6-[1-hydroxy-(1S)-pentyl]-4-methoxy-(6S)-2H, 5H-pyran-2-one, LL-P880β, pestalrone A, and methyl (E)-octadec-9-enoate, were isolated from the fermentation broth of *Pestalotiopsis* sp. YMF1.0474. The first four compounds showed inhibitory activity against acetylcholinesterase (AChE). The IC50 values were 33.90, 81.54, 16.43 and 95.22 μM, respectively [39]. Three bioactive benzophenone derivatives and five ambuic acid analogs were isolated from the fermentation broth of *Pestalotiopsis* sp., among which Pestalachlorides G is a rare natural dichlorobenzophenone derivative. It has good antibacterial activity against *Escherichia coli*, *Pseudomonas aeruginosa*, *Staphylococcus aureus*, *Bacillus subtilis*, *Candida glabrata*, etc. [40]. Two 2 H-pyranones and two isocoumarins were isolated from the solid culture of *P. microspora* SC3082 [41]. A new azaphilonoid, pitholide E, was isolated from the fermentation broth of *P. microspora* and ethyl acetate extract of mycelia. In addition, eight metabolites, pitholide B, pitholide D, pestalotin (LL-P880α), PC-2, LL-P880β, tyroso, and 4-oxo-4H-pyran-3-acetic acid, were isolated. These extracts have the same resistance to *C. cladosporioides*. Twelve polyketide derivatives and seven known analogs of heterocornols A-L were isolated from *P. heterocornis* cultures. The pathogenic active component of *Pestalotiopsis* insignis is a nitrogen-containing plant polysaccharide [42]. However, the metabolites produced by *Pestalotiopsis kenyana* have not been reported.

Pathogenic fungi can produce toxic metabolites in their hosts, which is considered to be one of the main causes of plant diseases. Usually, pathogenic fungi pathogenic factors mainly include three categories: enzymes, hormones, and toxins. Many pathogens can produce a series of enzymes that degrade the host cell wall and cell membrane components. These enzymes are generally induced and can be detected in infected tissues. Cell wall degrading enzymes can degrade the cell wall and cuticle of host plants, which is conducive to the invasion, colonization, and expansion of pathogenic fungi. These enzymes mainly include pectinase, cellulase, hemicellulase, protease, and phospholipase [43]. *Fusarium graminearum* secretes cell wall degrading enzymes such as cellulase, xylanase, and pectinase in the process of infection and expansion, resulting in the decomposition of the host cell wall components and cell wall relaxation, which is conducive to the pathogen infection and expansion in the host spike tissue [44]. After inoculation of *Ustilago maydis* in maize, the cell wall degrading enzyme activity produced by the virulent strain was higher, while that in the weak virulent strain and healthy maize tissue was much lower [45]. Hormones are synthesized by pathogens and are similar to or the same as plant growth regulators, which can disrupt the normal physiological metabolism of the host and cause growth deformity and lesions of the host. *Botrytis cinerea* can produce a large amount of abscisic acid, ethylene, and other plant hormones involved in its pathogenic process [46]. Toxins are closely related to the course of the disease and have been studied most. They are the products of mutual recognition and interaction between pathogens and plants and are toxic substances that play a decisive role in the course of plant disease [47]. Mycotoxins are mostly low molecular weight secondary metabolites, often composed of peptides or proteins, sugars, lipids, aromatic rings, hand eterocyclic, and organic acids. Toxins are closely related to the course of the disease and have been studied most. They are the products of mutual recognition and interaction between pathogens and plants and are toxic substances that play a decisive role in the course of plant disease [48]. *Fusarium* toxins are a variety of toxic metabolites produced during the growth of *Fusarium*, mainly including zearalenone, trichothecenes, fumonisins, and fumonisins, which can cause plant wilt, root rot, ear rot, and other rot diseases [49]. *Pestalotiopsis kenyana* causes a large number of infections in pepper leaves, and whether metabolites play a major pathogenic role should be confirmed.

In this experiment, LC-MS technology was used for full spectrum analysis of the samples to obtain MS and MS2 data [50,51]. The original data were preprocessed using Compound Discoverer 3.1 data processing software. Firstly, through the simple screening of parameters such as retention time and mass-to-charge ratio, peak alignment was performed for different samples according to retention time deviation and mass deviation (parts per million, ppm). Peak extraction was then performed according to the set ppm, signal-to-noise ratio (S/N), adduct ions, and other information, and the peak area was quantified. The metabolites were then identified by comparing the high-resolution secondary spectrum databases mzCloud and mzVault, as well as the MassList primary database search. The pathogen of *Z. schinifolium* leaf spot was isolated in the early stage. This experiment enabled the exploring of its pathogenic mechanism, analyzing the whole metabolome of the pathogen in three stages and the pathogen cultured on two media, and determining the metabolites that make the plant pathogenic through the analysis of differential metabolites, thereby providing a theoretical basis for the prevention and treatment of *Z. schinifolium* leaf spot.

## 2. Materials and Methods

### 2.1. Fungal Samples and Plant Samples

Fungal samples: *P. kenyana* was isolated from *Z. schinifolium*, and was provided by the forest protection laboratory of Sichuan Agricultural University [12]. (GenBank accession, ITS: NR147549.1, LSU: MH870724.1, PRB2: MH554958.1, TUB: KM199395.1, TEF: KM199502.1)

Plant samples: One year-old *Z. schinifolium* was purchased from Meishan *Z. schinifolium* nursery, with a plant height of about 30–40 cm, was planted in Sichuan Agricultural University Chengdu Campus fifth teaching building in a greenhouse (temperature: 25 °C, relative humidity: 55%).

### 2.2. Instruments and Reagents

A Q Exactive TM HF, Thermo Fisher mass spectrometer (Waltham, MA, USA), Vanquish UHPLC, Thermo Fisher chromatograph (Waltham, MA, USA), Hypesil Gold chromatographic column (100 × 2.1 mm, 1.9 μm), Thermo Fisher (Waltham, MA, USA), and a Thermo Fisher D3024R Scilogex cryogenic centrifuge (Waltham, MA, USA) were used.

Methanol (Thermo Fisher), formic acid (Thermo Fisher), ammonium acetate (Thermo Fisher), l-phenylalanine (Bio-BBI; Shanghai, China), *N*-acetyl-l-phenylalanine (Bio-BBI; Shanghai, China), 3-nitro-l-tyrosine (Bio-BBI; Shanghai, China), 2-(1*H*-indol-3-yl) acetic acid (Xilong; Guangzhou, China), *S*-(5-adenosy)-l-homocysteine (Kelong; Chengdu, China), 3,5-dimethoxybenzoic acid (Kelong; Chengdu, China), *N*-Acetylhistidine (Kelong; Chengdu, China), 2-aminobenzenesulfonic acid (Kelong; Chengdu, China), l-tyrosine (Bio-BBI; Shanghai, China), l-glutamic acid (Kelong; Chengdu, China), dl-Lysine (Bio-BBI; Shanghai, China), 3-methyl-2-oxobutanoic acid (Solarbio; Beijing, China), glycyl-l-leucine (Kelong; Chengdu, China), 3-hydroxy-3-methylpentanedioic acid (Ruiyong; China), dl-tryptophan (BBI; Shanghai, China), dl-arginine (Kelong; Chengdu, China), and 2-(2-acetyl-3,5-dihydroxyphenyl) acetic acid (Kelong; Chengdu, China) were used as reagents.

### 2.3. Sample Preparation before Processing

The pathogens were inoculated in a potato glucose medium at 25 °C, and the growth of pathogens was observed every day to determine the time of sample collection. Samples were collected at 3, 6, and 15 days. A total of six groups of samples were tested, with one biological duplicate sample in each group, split into six samples for technical repetition, yielding a total of 41 samples (Table 1). In order to control the quality of this experiment, QC samples were prepared at the same time to balance the status of the chromatography–mass spectrometry system and monitoring instruments, and the stability of the system was evaluated throughout the experiment. A blank sample was also set, primarily to remove background ions.

### 2.4. Metabolite Extraction

The sample was placed in an EP tube, and 300 μL of 80% methanol aqueous solution was added. The sample was frozen in liquid nitrogen for 5 min; after melting on ice, it was vortexed for 30 s, ultrasonicated for 6 min, and centrifuged at 5000 rpm at 4 °C for 1 min. The supernatant was placed in a new centrifuge tube and freeze-dried to a dry powder. According to the volume of the sample, the corresponding 10% methanol solution was added for solvation before injection into LC–MS for analysis.

### 2.5. LC-MS/MS Analysis

#### 2.5.1. Chromatographic Separation

Throughout the analysis process, the samples were placed in an automatic sampler at 4 °C, and the samples were separated using a Vanquish UHPLC ultrahigh liquid chromatography system (Thermo Fisher) with an HILIC column. The injection volume was 2 μL, the column temperature was 40 °C, and the flow rate was 0.2 mL/min (positive mode mobile phase: A, 0.1% formic acid; B, methanol; negative mode mobile phase: A, 5 mM ammonium acetate, pH 9.0; B, methanol). The chromatographic gradient elution procedure was as follows: 0–1.5 min, 98% A; 1.5–12 min, 98% A linearly decreased to 0% A; 12–14 min, 0% A linearly increased to 98% A; 14.1–17.0 min, A maintained at 98%. QC samples were inserted into the sample queue to detect and evaluate the stability of the system and the reliability of the experimental data.

#### 2.5.2. Mass Spectrometry Acquisition

Each sample was detected by electrospray ionization (ESI) in positive and negative ion modes. The samples were separated by UPLC and analyzed using a Q ExactiveTM HF mass spectrometer (Thermo Fisher). The ESI source conditions were as follows: spray voltage, 3.2 kV; sheath gas flow rate, 40 arb; auxiliary gas flow rate, 10 arb; capillary temperature, 320 °C; polarity, positive or negative. MS/MS secondary scans were data-dependent scans.

### 2.6. Pathogenicity Detection

According to the analysis of metabolomics data, a total of 18 possible pathogenic substances were selected: l-phenylalanine, *N*-acetyl-l-phenylalanine, 3-nitro-l-tyrosine, l-tyrosine, 2-(2-acetyl-3,5-dihydroxy phenyl)acetic acid, *S*-(5-adenosy)-l-homocysteine, 3,5-dimethoxybenzoic acid, *N*-acetylhistidine, 2-aminobenzenesulfonic acid, l-glutamic acid, dl-lysine, 3-methyl-2-oxobutanoic acid, glycyl-l-leucine, 3-hydroxy-3-methyl pentane dioic acid, dl-tryptophan, and dl-arginine, 2-(1*H*-indol-3-yl)acetic acid. Standard products were purchased for pathogenicity testing. 

The leaves of *Z. schinifolium* with uniform growth and no pests and diseases were disinfected with 75% alcohol and washed with sterile water three times. The above reagents were configured into solutions with concentrations of 20 μg/mL, 40 μg/mL, 60 μg/mL, 80 μg/mL, and 100 μg/mL, and then sprayed on the leaves of *Z. schinifolium*, 1 mL per leaf, with sterile water and 1 × 10^6^ cfu/mL pathogen spore suspension used as controls. The treated leaves were placed in a petri dish with wet filter paper, one leaf per dish, with moisturizing culture at 25 °C. Each treatment was repeated 10 times to observe the incidence and record the percentage of the lesion area to the total leaf area. The grid method was used to calculate the area, and the images of *Z. schinifolium* leaf spot obtained through proportional scanning were printed on the grid paper with the known area (0.04 cm^2^/grid). The number of grids occupied by leaves and lesions was counted, and the area of leaves and lesions was determined.

### 2.7. Data Processing

For multivariate statistical analysis, metaX was used to convert the data and then PLS-DA was performed to obtain the VIP value of each metabolite [52]. For univariate analysis, the statistical significance (*p*-value) of each metabolite between the two groups was calculated according to the *t*-test, and the fold change (FC value) of the metabolites between the two groups was calculated. The default criteria for screening differential metabolites were as follows: VIP > 1, *p* < 0.05, and 2 ≤ FC ≤ 0.5.

## 3. Results

### 3.1. Experimental Quality Evaluation

#### 3.1.1. QC Sample Correlation Analysis

It can be seen from Figure 1 that the *R^2^* of QC samples in this experiment was close to 1, i.e., the correlation of QC samples was high, indicating that the whole detection process had good stability and high data quality.

#### 3.1.2. Overall Sample PLS-DA

The overall sample was analyzed using the PLS-DA model, and the results are shown in Figure 2. The experimental samples in the positive and negative ion modes were slightly discrete but within the 95% confidence interval. The samples of the two media in the same culture stage overlapped slightly, and the separation between groups in each culture stage was more obvious, indicating the good repeatability of this experiment.

### 3.2. PLS-DA for Each Comparison Group

PLS-DA model analysis was performed on the samples of each comparison group, and the results are shown in Figure 3 and Figure 4. It can be seen from the score plot that the samples in the positive and negative ion modes were obviously distinct. The *R^2^Y* and *Q^2^Y* values of each comparison group approached 1 in the positive and negative ion modes, with *R^2^Y* being greater than *Q^2^Y* (Table 2). Therefore, the model was stable and reliable under PLS-DA treatment.

### 3.3. Univariate Statistical Analysis

Appendix A presents the differential metabolites of positive and negative ion peak patterns under univariate statistics. The results of the univariate statistical analysis are shown in Figure 5 and Figure 6. It was preliminarily judged that the differential metabolites of pathogenic bacteria cultured at three stages on two media changed greatly.

### 3.4. Significantly Different Metabolites

Appendix A shows that, in positive and negative ion modes, the fold change range of metabolite content was 0.0073–475.5961, wherein the fold change was mainly higher for esters, acids, ketones, and alkaloids, such as LDGTS 18:1, laudanosine, and 3,3-dimethyl-5-oxo-5-(2-piperidinoanilino) pentanoic acid. There were 3922 differential metabolites in nine groups under positive and negative ion modes, including lipids and lipid molecules, organic acids and their derivatives, organic heterocyclic compounds, organic oxygen compounds, carbohydrate polyketides, nucleosides, nucleotides, and analogs. There were 24 common differential metabolites in PKP3 vs. PKP6, PKP3 vs. PKP15, and PKP6 vs. PKP15. There were 240 different metabolites in PKP3 vs. PKP6 and PKP3 vs. PKP15. There were 138 different metabolites in PKP3 vs. PKP6 and PKP6 vs. PKP15. A total of 41 differential metabolites were identical between PKP3 vs. PKP15 and PKP6 vs. PKP15. There were 51 differential metabolites in PKZ3 vs. PKZ6, PKZ3 vs. PKZ15, and PKZ6 vs. PKZ15. There were 207 differential metabolites in PKZ3 vs. PKZ6 and PKZ3 vs. PKZ15. There were 103 differential metabolites in PKZ3 vs. PKZ6 and PKZ6 vs. PKZ15. There were 322 differential metabolites in PKZ3 vs. PKZ15 and PKZ6 vs. PKZ15.

There were 400 differential metabolites in the PKP3 vs. PKZ3 group. The metabolites with higher changes in this group were HPK, cytosine, 5α-tetrahydrocortisol, etc. There were 395 differential metabolites in the PKP6 vs. PKZ6 group. The metabolites with higher changes in this group were 6,8-di(*tert*-butyl)-4-oxo-4*H*-chromene-2-carboxylic acid, gibberellin A4, coenzyme Q1, etc. There were 356 differential metabolites in the PKP15 vs. PKZ15 group. The differential metabolites in this group were xanthosine, 2′-*O*-methyladenosine *N*^2^, *N*^2^-dimethylguanosine, 4-hydroxytamoxifen, etc. There were 105 differential metabolites in the PKP3 vs. PKP6 group. The metabolites with higher fold changes in this group were MPK, LPA 18:3, rutarin, etc. There were 505 differential metabolites in the PKP3 vs. PKP15 group. The metabolites with higher fold changes in this group were laudanosine and 13,14-dihydro-15-keto-PGD2,5-methyl-8-nitro-3-spirocyclohexhex.

### 3.5. Hierarchical Cluster Analysis

A hierarchical cluster analysis was performed on the two groups of differential metabolites obtained, and the differences in metabolic expression patterns between the two groups and within the group were obtained. The relationship between the clustering of metabolite content between groups can be seen through horizontal comparison. The metabolites clustered in the same cluster had similar expression patterns, possibly reflecting a relatively close reaction step in the metabolic process.

It can be seen from Figure 7 and Figure 8 that the upregulated metabolites in different treatments of the samples were in the same cluster, while the downregulated metabolites were in another cluster. Metabolites of PKP3 vs. PKZ3, PKP6 vs. PKZ6, PKP15 vs. PKZ15, PKP3 vs. PKP6, PKP3 vs. PKP15, PKP6 vs. PKP15, PKZ3 vs. PKZ6, PKZ3 vs. PKZ15, and PKZ6 vs. PKZ15 were significantly different among different treatments. In the positive ion peak mode of the PKP3 vs. PKP3 group, the cluster analysis divided the metabolites into two categories. In PKP3, the metabolites from sulforidazine to 4-fluoro-*N*-[4-(4-methyl piperazino) phenyl] benzene sulfonamide were blue, indicating downregulation, belonging to the same cluster. In PKP6, these metabolites were red, indicating upregulation. The PKP3 vs. PKP3, PKP6 vs. PKZ6, PKP15 vs. PKZ15, PKP3 vs. PKP15, PKP6 vs. PKP15, PKZ3 vs. PKZ6, PKZ3 vs. PKZ15, PKZ6 vs. PKZ15, and PKZ6 vs. PKZ15 groups also had the same pattern. It can be seen that different medium formulations changed the metabolites of pathogenic bacteria, and the types of metabolites were different at different growth stages of pathogenic bacteria.

### 3.6. KEGG Metabolic Pathway Analysis

The significant differential metabolites obtained from each comparison group were subjected to KEGG ID mapping and submitted to the KEGG website for related pathway analysis using Metabo Analyst online to obtain the enrichment analysis of Figure 6. There were 369 metabolic pathways in cation mode and 263 metabolic pathways in anion mode. The top 20 metabolic enrichment pathways were sorted, and the enrichment analysis is shown in Appendix A.

A comparative analysis of the top 20 metabolic pathways involved in differential metabolites is shown in Figure 9. There were 14 differential metabolic pathways in the positive ion mode, and 13 differential metabolic pathways in the negative ion mode. Among them, there were many metabolites in the global and overview maps, the amino-acid metabolism, and the carbohydrate metabolism.

### 3.7. Pathogenicity Test Results

The *Z. schinifolium* leaves were treated with various reagents (20 μg/mL, 40 μg/mL, 60 μg/mL, and 80 μg/mL, 100 μg/mL). The leaves of 5-dimethoxybenzoic acid, *S*-(5-adenosy)-l-homocysteine, 2-(1*H*-indol-3-yl) acetic acid, l-glutamic acid, and 2-(2-acetyl-3,5-dihydroxy phenyl) acetic acid showed different degrees of yellowish-brown lesions, while CK_1_ did not show lesions (Figure 10). When the concentration was 20 μg/mL, 40 μg/mL, 80 μg/mL, and 100 μg/mL, there was no significant difference in the lesion area treated with 3,5-dimethoxybenzoic acid and *S*-(5-adenosy)-l-homocysteine. However, there were significant differences when treated with 2-(1*H*-indol-3-yl) acetic acid, l-glutamic acid, and 2-(2-acetyl-3,5-dihydroxy phenyl) acetic acid. The lesion area treated with 3,5-dimethoxybenzoic acid was significantly different from that of *S*-(5-adenosy)-l-homocysteine, 2-(1*H*-indol-3-yl) acetic acid, l-glutamic acid, and 2-(2-acetyl-3,5-dihydroxy phenyl) acetic acid at 60 μg/mL. From the perspective of reagent concentration, the lesion area treated with each reagent increased with the increase in concentration. Compared with the 1 × 10^6^ cfu/mL pathogen spore suspension treatment, the lesion area was reduced except for treatments with 100 μg/mL 3,5-dimethoxybenzoic acid and *S*-(5-adenosy)-l-homocysteine (Table 3).

## 4. Discussion

In addition to genetic genes, the external environment is also one of the important factors affecting the growth and physiological functions of organisms. In this experiment, the metabolomics of *P. kenyana* in the three stages of the mycelial logarithmic growth phase, mycelial growth stationary phase, and large spore phase and PDA, green pepper leaf induction culture conditions were analyzed, and more differential metabolites were obtained. Metabolomics analysis and hierarchical clustering results showed that the metabolic reactions induced by the pathogen green pepper leaves were more complex and the differential metabolites were more abundant. Therefore, during the growth and development of *P. kenyana*, the expression of metabolites closely related to the environment must be different in different environments. When *Rhizoctonia solani* was induced by leaf, sheath, and root, the metabolites in the three groups were significantly different. This study also found that the pathogenicity-related metabolites of fungi cultured on PDA containing the host plant *Z. schinifolium* may be increased compared to pathogens cultured on PDA. The up-regulated differential metabolites of pathogens induced by green pepper leaves mainly included organic acids, amino acids, and esters. Previous studies have found that organic acids are common metabolites of pathogens, and some organic acids can promote the growth of pathogens. When the content of malic acid and palmitic acid increased, the growth rate of *Fusarium oxysporum* increased significantly [53,54,55]. Nutrients are the basic conditions for the growth and reproduction of fungi, including water, carbon sources, nitrogen sources, inorganic salts, and necessary growth factors. As a component of proteins, amino acids are directly involved in a variety of physiological and biochemical processes in pathogen cells. Some amino acids are also directly used as nutrients in pathogens, or as functional sites for binding or modification between proteins. In addition to supporting cell basal metabolism, many amino acids can trigger virulence-related metabolic reactions and have a direct impact on the pathogenicity of fungi, such as hyphal morphogenesis and biofilm growth of Candida albicans, neospore capsule formation, and blackening of Aspergillus fumigatus [56]. The effector protein is a pathogenic factor secreted by pathogenic fungi during the infection of plant hosts and plays a vital role in pathogenesis. For example, small secreted cysteine-rich proteins (SCRPs) are common effectors that interact with hosts to manipulate host immune responses [57,58].

In addition, the growth stages of pathogens are different, and the types of metabolites are also different. The up-regulated differential metabolites of pathogenic bacteria in the large spore stage mainly include organic acids, amino acids, and other acids. Studies have shown that many plant pathogens’ metabolites contain organic acids, amino acids, etc. [59]. Botrytis cinerea is the main pathogen of gray mold, an important crop disease. It has a complex metabolic system. Its metabolites are mainly amino acids, alcohols, organic acids, sugars, etc., including toxins, cell wall degrading enzymes, abscisic acid, polysaccharides, laccase, etc. [60,61,62,63]. The m-hydroxyphenyl acetic acid in *Rhizoctonia solani* was identified as its pathogenic substance, which could inhibit the growth of beetroot at a concentration of more than 0.025% and produce necrotic spots at a concentration of more than 0.1% [64,65]. The results of metabolome analysis of *Magnaporthe oryzae* showed that the contents of most metabolites in the mycelia and appressoria of autophagic mutant Moatg1 decreased compared with wild-type GuyI1. Differential metabolites mainly include sugars, amino acids, polyols, and lipids, among which trehalose, isoleucine, valine, and glycerol have been reported to be closely related to the pathogenicity of *M. oryzae* [66,67]. Combined with previous research results, we believe that the differential metabolites such as organic acids and amino acids in this experiment may be related to the pathogenicity of *P. kenyana*.

KEGG metabolic pathway analysis showed that the induction of green pepper leaves mainly caused changes in metabolic pathways such as carbohydrates and amino acids. Carbohydrates in higher plants are mainly sugars, especially in the form of sucrose transported from the source to non-photosynthetic sink tissue. Carbon is the skeleton of bacteria and the source of energy. Glucose, sucrose, and starch are the most commonly used carbon sources for fungi. Sugars will eventually be exported transporters (SWEET) that can transport a variety of monosaccharides and sucrose, and play an important regulatory role in plant reproductive development and host-pathogen interaction [68]. The expression levels of three hexose transporters BcHXT1, BcHXT6, and BcHXT13 increased significantly after tomato leaves were infected by *Botrytis cinerea* [69]. Previous studies have found that carbohydrates can promote the growth of a variety of pathogenic fungi. Pathogens extract sugars from host cells as their nutrients and substrates to enhance pathogenicity. Carbohydrates can be used as carbon sources for the mycelial growth of *Fusarium solani* and promote spore germination. Aspartic acid can be used as a nitrogen source, and long-chain organic acids such as benzoic acid have low promotion and high inhibition effects on pathogens [70]. In addition, sugars can also affect the pathogenicity of pathogens by regulating transport-related proteins. Compared with the wild type, there were significant differences in the metabolism of organic acids and amino acids in the metabolic group. The difference in pathogenicity of *Bipolaris sorokiniana* may be related to the metabolism of organic acids and amino acids [71,72]. The differential expression pathways of attenuated and virulent strains of *Botrytis cinerea* mainly involve carbohydrate metabolism, lipid metabolism, amino acid metabolism, and the biosynthesis and metabolism of polysaccharides [73]. The above results are similar to the results of this experiment. Therefore, we believe that differential metabolites such as sugars, organic acids, and amino acids in this experiment may be related to the pathogenicity of *P. kenyana*.

According to the results of metabolomics analysis, some differential metabolites were selected for pathogenicity determination to determine the possible pathogenic substances. The results of pathogenicity experiments showed that 3,5-Dimethoxy benzoic acid, S-(5-Adenosy)-L-Homocysteine, 2-(1H-indol-3-yl) acetic acid, L-Glutamic acid, 2-(2-acetyl-3,5-dihydroxy phenyl) acetic acid could cause leaf disease after infecting the leaves of *Z. schinifolium*. Symptoms are most severe in 5-Dimethoxy benzoic acid and S-(5-Adenosy)-L-Homocysteine, which are similar to the natural state. P-hydroxybenzoic acid and methyl p-hydroxybenzoate have been isolated from rice sheath spot pathogen (*Rhizoctonia oryzae*), and p-hydroxybenzoic acid was screened as a phytotoxin [74]. Aromatic acid toxins with benzene rings were isolated from *R. solani*, including phenylacetic acid, hydroxyphenyl acetic acid, and hydroxybenzoic acid. The results of this experiment are similar to the above substance types, presumably due to the different types of pathogens, resulting in differences in the structure of their pathogenic metabolites.

In summary, the metabolites of leaf spot fungi were significantly different under the induction of *Z. schinifolium* leaves. Through the pathogenicity test, 3,5-dimethoxybenzoic acid and *S*-(5-adenosy)-l-homocysteine were screened as the most likely pathogenic substances, which provided a theoretical reference value for exploring the pathogenic mechanism of the pathogen. In the next step, on the basis of the experimental results, the metabolites of the pathogen of *Z. schinifolium* leaf spot should be isolated and purified, and their properties should be studied. Bioinformatics analysis and gene verification should be carried out on the genes corresponding to the related metabolites, so as to provide a theoretical basis for further study on the pathogenesis of *Z. schinifolium* leaf spot caused by *P. kenyana*, contributing to further prevention and control.

## Figures and Tables

**Figure 1 jof-08-01208-f001:**
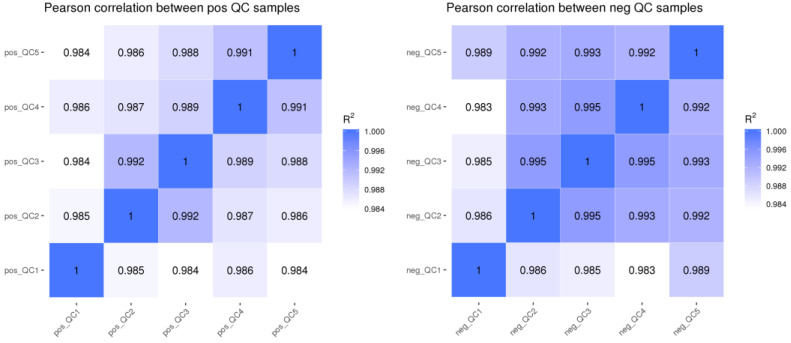
Correlation analysis of QC samples. Note: QC_1–5_ represents five replicates of QC samples. POS: positive ion mode; NEG: negative ion mode.

**Figure 2 jof-08-01208-f002:**
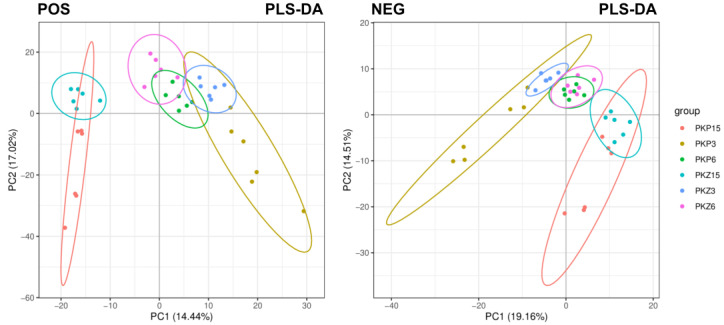
PLS-DA score plot of total samples in positive and negative ion modes. PC1: principal component 1; PC2: principal component 2. Scatter points of different colors represent samples of different experimental groups; the ellipse is the 95% confidence interval. PKP3–PKZ15 is the same as in Table 1. POS: positive ion mode; NEG: negative ion mode.

**Figure 3 jof-08-01208-f003:**
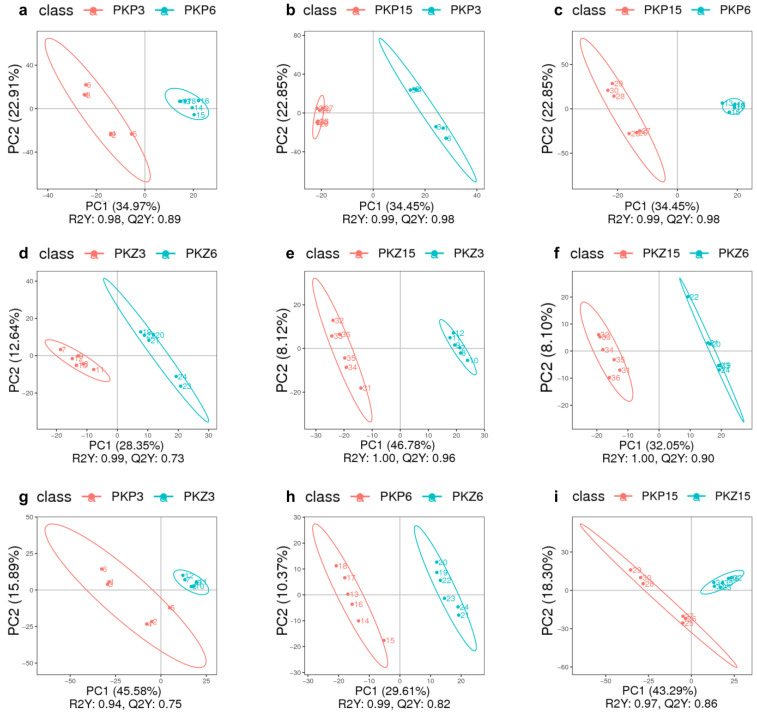
PLS-DA score plot of positive and negative ion modes for each comparison group (positive ion mode). PKP3–PKZ15 is the same as Table 1. (**a**): PLS-DA score plot between PKP3 and PKP6; (**b**): PLS-DA score plot between PKP3 and PKP15; (**c**): PLS-DA score plot between PKP6 and PKP15; (**d**): PLS-DA score plot between PKZ3 and PKZ6; (**e**): PLS-DA score plot between PKZ3 and PKZ15; (**f**): PLS-DA score plot between PKZ6 and PKZ15; (**g**): PLS-DA score plot between PKP3 and PKZ3; (**h**): PLS-DA score plot between PKP6 and PKZ6; (**i**): PLS-DA score plot between PKP15 and PKZ15.

**Figure 4 jof-08-01208-f004:**
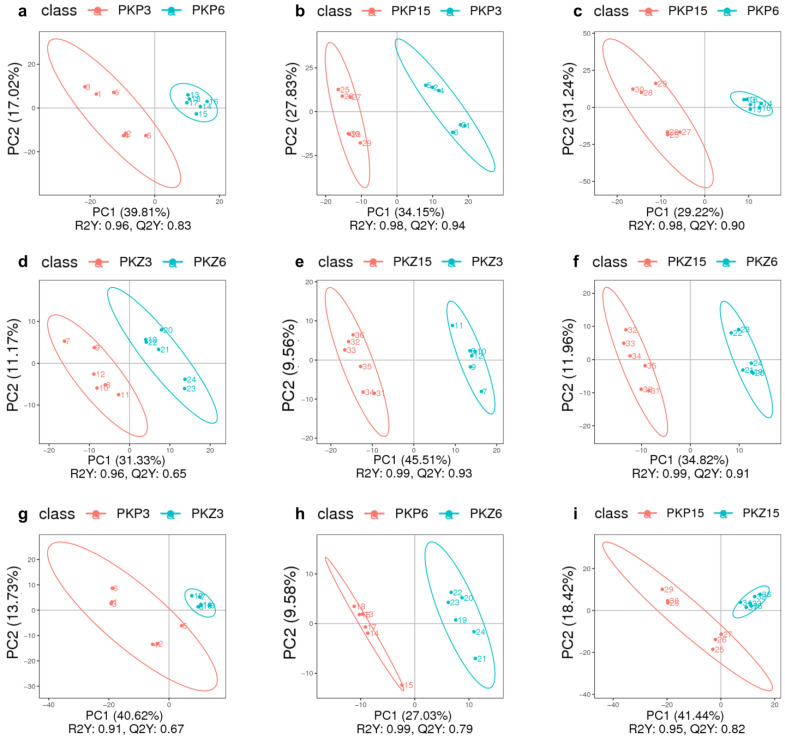
PLS-DA score plot of positive and negative ion modes for each comparison group (negative ion mode). PKP3–PKZ15 is the same as Table 1. (**a**–**i**) is the same as Figure 3.

**Figure 5 jof-08-01208-f005:**
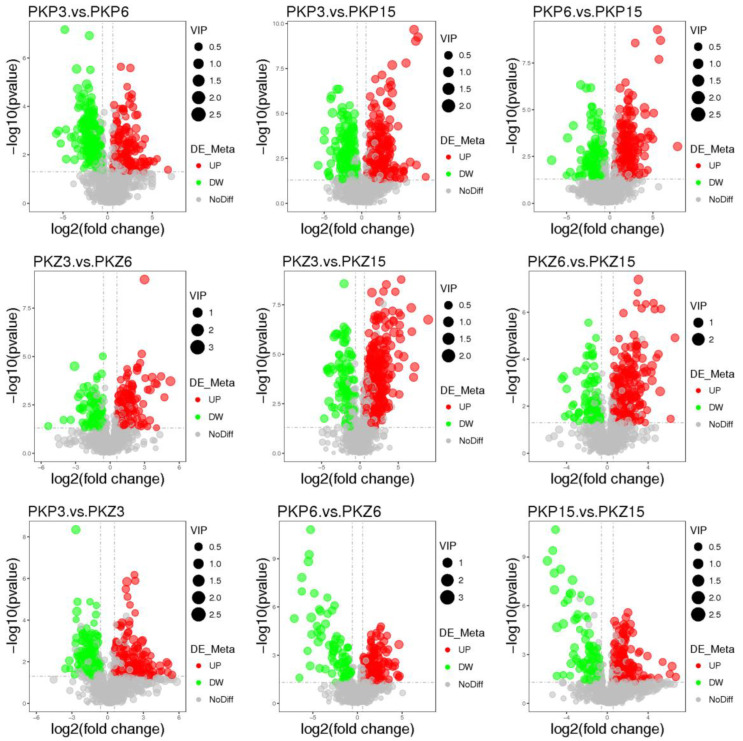
PLS-DA score plot of positive and negative ion modes for each comparison group (positive ion mode). The horizontal axis represents the log_2_ fold change of the metabolite in different groups, the vertical axis represents the different significance levels (−log_10_
*p*-value), and each point in the volcano plot represents a metabolite. Gray dots indicate metabolites with no significant change. Red dots represent significantly upregulated metabolites, green dots represent significantly downregulated metabolites, and the dot size represents the VIP value. PKP3–PKZ15 is the same as in Table 1.

**Figure 6 jof-08-01208-f006:**
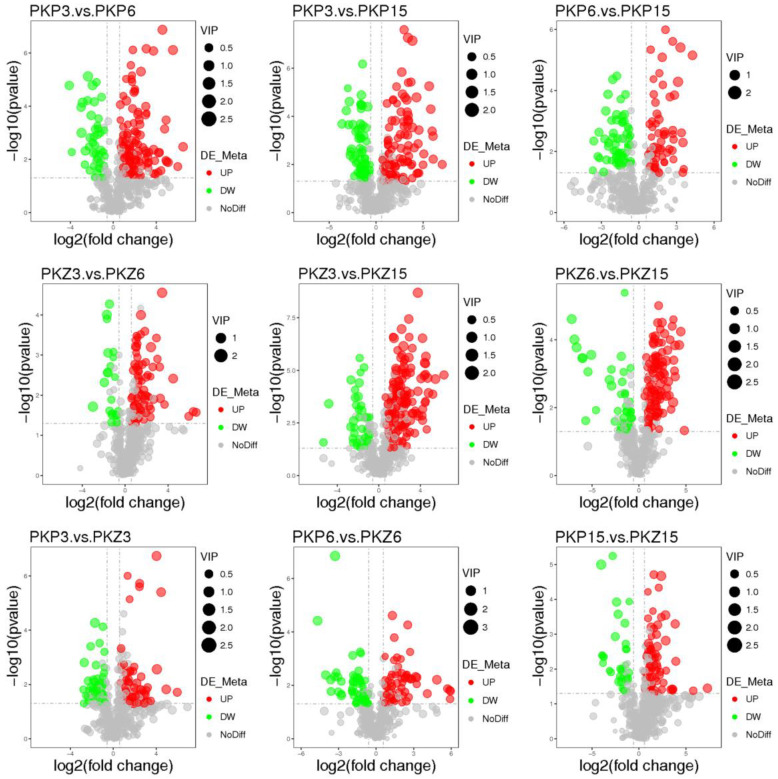
PLS-DA score plot of positive and negative ion modes for each comparison group (negative ion mode). PKP3–PKZ15 is the same as Table 1.

**Figure 7 jof-08-01208-f007:**
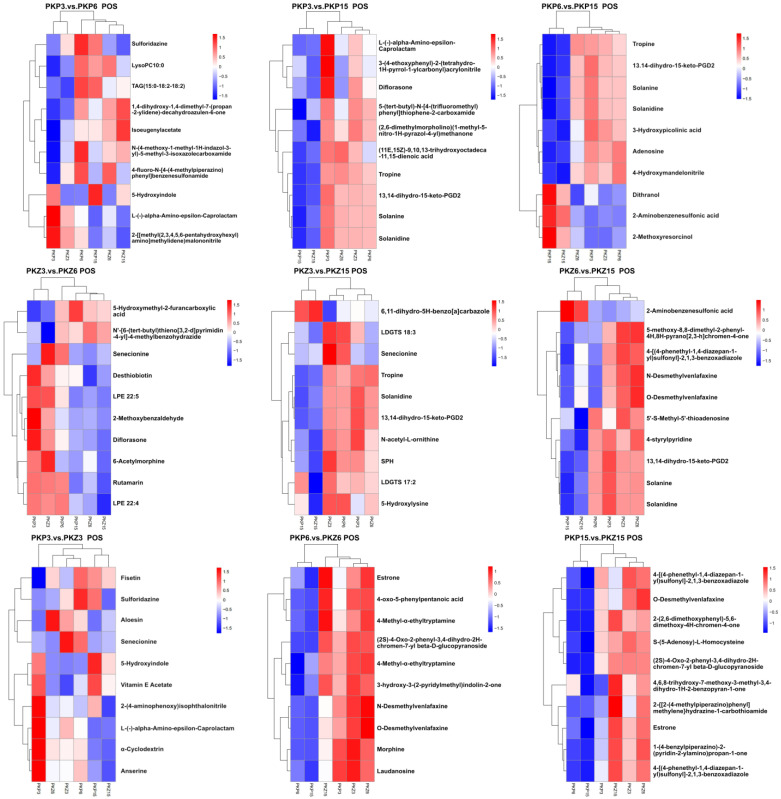
Hierarchical clustering results of significant differential metabolites (top 10 in each comparison group, positive ion mode). The figure shows the results of the hierarchical clustering of metabolites within and between groups (red indicates the upregulation of metabolites, and blue indicates downregulation). The longitudinal axis is the clustering of samples, and the transverse axis is the clustering of metabolites. A shorter clustering branch indicates higher similarity. PKP3–PKZ15 is the same as in Table 1. POS: positive ion mode.

**Figure 8 jof-08-01208-f008:**
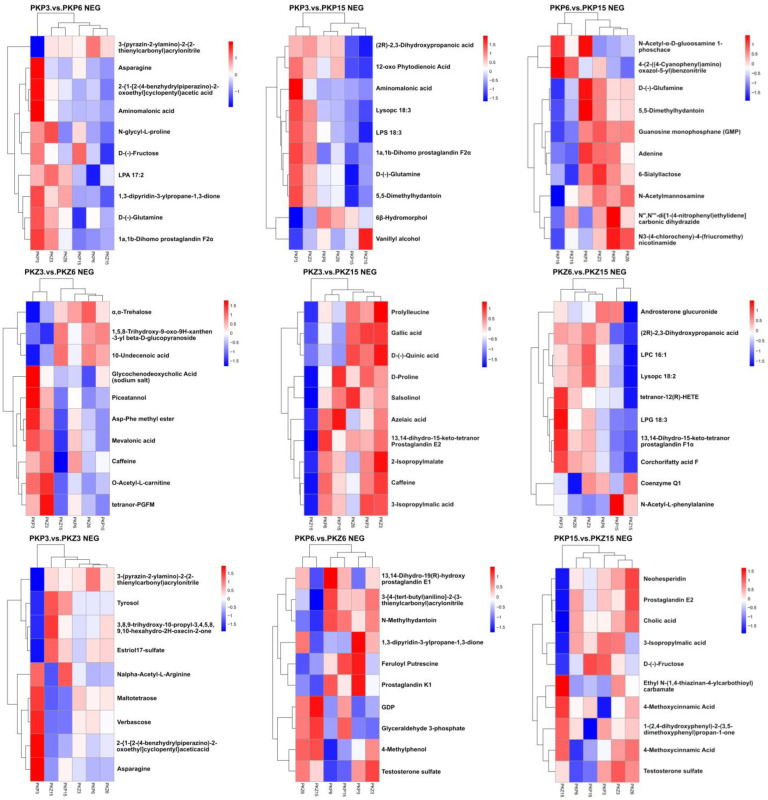
Hierarchical clustering results of significant differential metabolites (top 10 in each comparison group, negative ion mode). NEG: negative ion mode. PKP3–PKZ15 is the same as in Table 1.

**Figure 9 jof-08-01208-f009:**
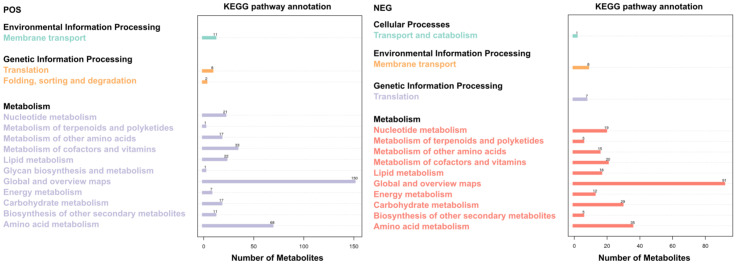
Significantly different metabolites are involved in metabolic pathways. The abscissa represents the number of metabolites, while the ordinate represents the annotated KEGG pathway; the diagram shows the number of metabolites annotated by each secondary classification under the primary classification of pathways. POS: positive ion mode; NEG: negative ion mode.

**Figure 10 jof-08-01208-f010:**
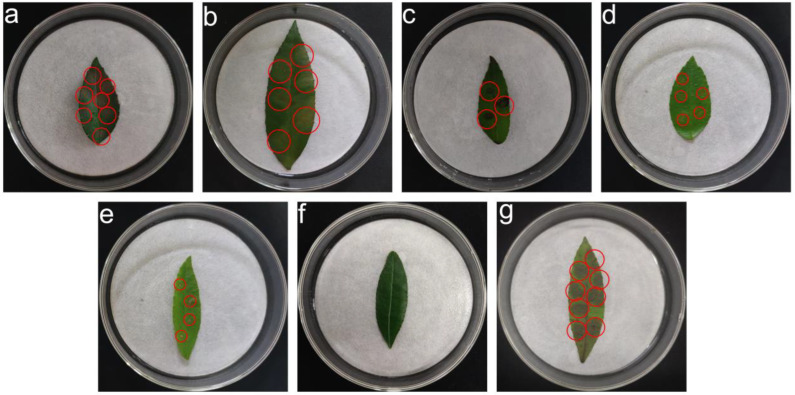
Pathogenicity test results of 100 μg/mL treatment. (**a**) 3,5-Dimethoxybenzoic acid; (**b**) *S*-(5-adenosy)-l-homocysteine; (**c**) 2-(1*H*-indol-3-yl)acetic acid; (**d**) l-glutamic acid; (**e**) 2-(2-acetyl-3,5-dihydroxy phenyl)acetic acid; (**f**) CK_1_: sterile water; (**g**) CK_2_ (1 × 10^6^ cfu/mL pathogen spore suspension is not photographed).The red circle marks the lesions.

**Table 1 jof-08-01208-t001:** Sample information.

Sample Group	Group Name	Number of Samples in the Group	Sample Status
**1**	PKP3	6	solid
**2**	PKZ3	6	solid
**3**	PKP6	6	solid
**4**	PKZ6	6	solid
**5**	PKP15	6	solid
**6**	PKZ15	6	solid
**7**	QC	5	liquid

Note: The first two letters (PK) in groups 1–6 represent *P. kenyana*, the third letter (P/Z) represents PDA or PDA + 10 g of *Z. schinifolium* leaf medium, and the number represents the culture time of the pathogen. For example, PKP3 was inoculated in PDA medium for 3 days. The QC samples had equal volumes to the experimental samples.

**Table 2 jof-08-01208-t002:** Evaluation parameters of the PLS-DA model.

Sample Comparison Group	Ion Mode	A	*R^2^Y* (cum)	*Q^2^Y* (cum)	*R^2^* Intercept	*Q^2^* Intercept
PKP3 vs. PKZ3	+	2	0.94	0.75	0.81	−0.92
PKP3 vs. PKZ3	−	2	0.91	0.67	0.76	−0.98
PKP6 vs. PKZ6	+	2	0.99	0.82	0.88	−0.88
PKP6 vs. PKZ6	−	2	0.99	0.79	0.89	−0.8
PKP15 vs. PKZ15	+	2	0.97	0.86	0.75	−0.89
PKP15 vs. PKZ15	−	2	0.95	0.82	0.74	−1.04
PKP3 vs. PKP6	+	2	0.98	0.89	0.81	−0.97
PKP3 vs. PKP6	−	2	0.96	0.83	0.78	−0.99
PKP3 vs. PKP15	+	2	0.99	0.98	0.73	−1.32
PKP3 vs. PKP15	−	2	0.98	0.94	0.71	−1.17
PKP6 vs. PKP15	+	2	0.99	0.96	0.8	−1.08
PKP6 vs. PKP15	−	2	0.98	0.9	0.78	−1.2
PKZ3 vs. PKZ6	+	2	0.99	0.73	0.87	−0.8
PKZ3 vs. PKZ6	−	2	0.96	0.65	0.85	−0.79
PKZ3 vs. PKZ15	+	2	1	0.96	0.84	−0.74
PKZ3 vs. PKZ15	−	2	0.99	0.93	0.81	−0.85
PKZ6 vs. PKZ15	+	2	1	0.9	0.89	−0.8
PKZ6 vs. PKZ15	−	2	0.99	0.91	0.84	0.81

Note: PKP3–PKZ15 is the same as Table 1.

**Table 3 jof-08-01208-t003:** Effects of different reagents and their concentrations on the lesion area of *Z. schinifolium*.

Reagent	Concentration (μg/mL)				
	20	40	60	80	100
3,5-Dimethoxybenzoic acid	17.08 ± 0.67 aE	22.85 ± 0.81 aD	27.25 ± 0.46 aC	33.43 ± 0.81 aB	38.15 ± 0.91 aA
*S*-(5-Adenosy)-l-homocysteine	16.25 ± 0.35 aE	21.83 ± 0.51 aD	24.63 ± 0.31 bC	32.70 ± 0.63 aB	37.13 ± 0.25aA
2-(1*H*-Indol-3-yl)acetic acid	3.10 ± 0.14 bD	3.48 ± 0.09 bC	4.23 ± 0.11 cB	4.50 ± 0.15 bB	5.48 ± 0.23 bA
l-Glutamic acid	5.83 ± 0.15 cD	6.80 ± 0.15 cC	8.05 ± 0.13 dB	8.50 ± 0.09 cB	10.53 ± 0.33 cA
2-(2-acetyl-3,5-dihydroxyphenyl)acetic acid	1.70 ± 0.13 dD	2.13 ± 0.06 dC	2.35 ± 0.06 eB	2.93 ± 0.09 dB	3.48 ± 0.18 dA
l-Phenylalanine	0.00 ± 0.00 eA	0.00 ± 0.00 eA	0.00 ± 0.00 fA	0.00 ± 0.00 eA	0.00 ± 0.00 eA
*N*-Acetyl-l-phenylalanine	0.00 ± 0.00 eA	0.00 ± 0.00 eA	0.00 ± 0.00 fA	0.00 ± 0.00 eA	0.00 ± 0.00 eA
3-Nitro-l-tyrosine	0.00 ± 0.00 eA	0.00 ± 0.00 eA	0.00 ± 0.00 fA	0.00 ± 0.00 eA	0.00 ± 0.00 eA
*N*-Acetylhistidine	0.00 ± 0.00 eA	0.00 ± 0.00 eA	0.00 ± 0.00 fA	0.00 ± 0.00 eA	0.00 ± 0.00 eA
2-Aminobenzenesulfonic acid	0.00 ± 0.00 eA	0.00 ± 0.00 eA	0.00 ± 0.00 fA	0.00 ± 0.00 eA	0.00 ± 0.00 eA
l-Tyrosine	0.00 ± 0.00 eA	0.00 ± 0.00 eA	0.00 ± 0.00 fA	0.00 ± 0.00 eA	0.00 ± 0.00 eA
dl-Lysine	0.00 ± 0.00 eA	0.00 ± 0.00 eA	0.00 ± 0.00 fA	0.00 ± 0.00 eA	0.00 ± 0.00 eA
3-Methyl-2-oxobutanoic acid	0.00 ± 0.00 eA	0.00 ± 0.00 eA	0.00 ± 0.00 fA	0.00 ± 0.00 eA	0.00 ± 0.00 eA
Glycyl-l-leucine	0.00 ± 0.00 eA	0.00 ± 0.00 eA	0.00 ± 0.00 fA	0.00 ± 0.00 eA	0.00 ± 0.00 eA
3-hydroxy-3-methyl pentane dioic acid	0.00 ± 0.00 eA	0.00 ± 0.00 eA	0.00 ± 0.00 fA	0.00 ± 0.00 eA	0.00 ± 0.00 eA
dl-Tryptophan	0.00 ± 0.00 eA	0.00 ± 0.00 eA	0.00 ± 0.00 fA	0.00 ± 0.00 eA	0.00 ± 0.00 eA
dl-Arginine	0.00 ± 0.00 eA	0.00 ± 0.00 eA	0.00 ± 0.00 fA	0.00 ± 0.00 eA	0.00 ± 0.00 eA
3,5-Dimethoxybenzoic acid	0.00 ± 0.00 eA	0.00 ± 0.00 eA	0.00 ± 0.00 fA	0.00 ± 0.00 eA	0.00 ± 0.00 eA
CK_1_	0.00 ± 0.00				
CK_2_	35.18 ± 1.89				

Note: CK_1_: sterile water, CK_2_: 1 × 10^6^ cfu/mL pathogen spore suspension. The data in the table are the percentage lesion area with respect to total leaf area (%), and the data in the table are the average of 10 replicates. Different lowercase letters indicate different reagents at the same concentration of *Z. schinifolium* lesion area at the level of *p* < 0.05 (least significant difference (LSD) method). Different capital letters indicate a difference in the area of *Z. schinifolium* lesions for the same reagent at different concentrations at *p* < 0.05 level (LSD method).

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
