# Peer review of "Differential Metabolomics Reveals Pathogenesis of Pestalotiopsis kenyana Causing Leaf Spot Disease of Zanthoxylum schinifolium"

_jof, 2022, doi:10.3390/jof8111208_

Round 1

Reviewer 1 Report (Previous Reviewer 1)

The manuscript has been greatly improved by the authors and so it is suitable for publication on the present form.

Reviewer 2 Report (Previous Reviewer 2)

This revised MS is significantly improved and the authors have addressed all my concerns and comments.

This manuscript is a resubmission of an earlier submission. The following is a list of the peer review reports and author responses from that submission.

Round 1

Reviewer 1 Report

The manuscript "Differential Metabolomics Reveals Pathogenesis of Pestalotiopsis kenyana Causing Leaf Spot Disease Induced by Zanthoxylum bungeanum" is about the use of a metabolomic approach to identify differential metabolites secreted by P. kenyana at different growth stages and in different media which may be involved in its pathogenesis. These results were supported by pathogenicity tests of pure compounds applied on healthy leaves. However, there are flaws throughout the whole manuscripts, mostly in results and in discussion section. For instance, there are no PCA score plots reported in the main text and in supplementary materials that are cited in the manuscript. Figure numbers do not correspond to those reported and described in the manuscript  (i.e., page 15, line 341, figure 6 is reported but the text describe figure 9). Discussion section appears to be kind of small compared to the results obtained and reported in the results section. In fact, the first part (lines 372-432) is unnecessary as it is like a second introduction, leaving very few lines for the experimental results. The discussion about the modulation of metabolic profiles is missing; lines 438-447 are just a summary of pathogenicity test results with no comments or even hypothesis supported by similar/opposite findings in literature data. For these reasons, the manuscript is nor suitable for publication in Journal of Fungi.

Reviewer 2 Report

This MS mainly explored the metabolites of the pathogen (P. kenyana) causing Z. schinifolium leaf spot with a LC-MS based metabolome analysis.  The results of this work could provide more information on the pathogenic metabolites and mechanism of Pkenyana. While this MS still needs significant improvement. This MS's current title is confusing (....induced by Zanthoxylum bungeanum???). The introduction of this MS is hardly followed by readers,  which has relatively flawed logic. Results need to pay more attention to the underlying biological meaning and objectives of the corresponding experiments rather than simply describing the obtained data without deep digging and analysis. Besides, this MS needs further editing for language and writing quality.